Do you even exercise, ref? Exploring habits of Spanish basketball referees during practice and matches

Suárez-Iglesias David 1 dsuai@unileon.es
González-Devesa Daniel 2
Ayán Carlos 3
Sánchez-Sixto Alberto 4 5
Vaquera Alejandro 1 6
1 Universidad de León, Faculty of Physical Activity and Sports Sciences, VALFIS Research Group, Institute of Biomedicine (IBIOMED) , León , Spain
2 Faculty of Physical Activity and Sports Sciences, Universidad de León , León , Spain
3 Departamento de Didácticas Especiais, Well-Move Research Group, Galicia Sur Health Research Institute (IIS Galicia Sur), SERGAS-UVIGO, Universidad de Vigo , Pontevedra, Galicia , Spain
4 Physical Performance & Sports Research Center, Universidad Pablo de Olavide , Sevilla, Andalucía , Spain
5 Department of Sport, CEU Cardenal Spínola , Bormujos, Andalucía , Spain
6 School of Sport & Exercise Science, University of Worcester , Worcester , United Kingdom
Chen Yung-Sheng
Electronic publication date: 2024 Jan 29
Publication date: 2024
Volume: 12
Electronic Location ID: e16742
Received 2023 Mar 21; Accepted 2023 Dec 11
Copyright: © 2024 Suárez-Iglesias et al.
Copyright year: 2024
Copyright holder: Suárez-Iglesias et al.
License: This is an open access article distributed under the terms of the Creative Commons Attribution License, which permits unrestricted use, distribution, reproduction and adaptation in any medium and for any purpose provided that it is properly attributed. For attribution, the original author(s), title, publication source (PeerJ) and either DOI or URL of the article must be cited.
License URL: https://creativecommons.org/licenses/by/4.0/

Keywords: Fitness, Performance, Strength, Team sport, Training

Funding: The authors received no specific funding for this work.

==============================
Background

Basketball referees are a vital part of the organised competition system, although they remain an “outgroup” in sport. While physical development and fitness programming are deemed necessary for basketball officiating excellence, there is a paucity of literature exploring strategies for physical fitness management in this population.

Methods

This research was a nationwide cross-sectional, self-administered online survey conducted in 2021. A sample of 628 (531 males, 97 females) referees from 18 regional referee organisations in Spain provided individual responses to gather information on demographic details, level of participation in refereeing, physical fitness practices, and match-day exercise-based regimens. The data were described using summary statistics, and the associations of the assessed variables were subsequently calculated using contingency tables.

Results

Our findings reveal that a large fraction of the Spanish basketball referee population focuses on aerobic (83%) and strength (73.6%) activities, while less attention is paid to speed (36.9%) and flexibility (23.2%), and agility, coordination, and balance tasks are somewhat overlooked. No significant differences were observed among the referee categories regarding weekly training days or session duration, with most training for 15–60 min per session. Elite referees were more likely to hire personal trainers and engage in strength and flexibility exercises. Sub-elite referees showed a higher tendency to perform stretching and joint mobility activities post-match, while regional referees did so less frequently. Approximately 30.7% of referees across all competitive levels engaged in re-warm-up (RW-U) activities, with stretching and joint mobility being the most prevalent.

Conclusions

Spanish basketball referees participate in routine physical exercise and fitness practices, irrespective of their competition level. While warm-up activities are prevalent, some sub-elite and regional referees do not consistently perform them, and re-warm-up routines are not extensively embraced.

Introduction

Referees are essential in maintaining professional sports’ integrity and legitimacy, as they operate in high-pressure environments and are expected to make unbiased judgments (Cunningham, Mergler & Wattie, 2022). This task is particularly challenging for basketball referees, who must make quick and accurate decisions under high-pressure situations (Karacam & Adiguzel, 2019). Changes in basketball rules have led to a more dynamic game with increased physical demands (García-Santos et al., 2020), which may result in heightened physiological fatigue during competition and hinder decision-making performance (Nabli et al., 2019). Consequently, physical fitness is considered a critical aspect of competence for basketball referees (Anshel, 1995), and poor physical condition has been found to negatively impact their decision-making accuracy (Nabli et al., 2016b). This highlights the importance of effective physical conditioning programs for both elite (Leicht et al., 2020) and sub-elite basketball referees (Leicht et al., 2019).

Nevertheless, scant research exists on how referees train their physical condition, despite existing recommendations in this regard (García-Santos et al., 2020), or what competition routines are usually put into practice by them. The study by de Paula, da Cunha & Andreoli (2021), unique in its focus on physical fitness practices among basketball referees, reported on 78 regional-level Brazilian referees. It specifically provided data for only three types of exercises—bodybuilding, running, and one other category—covering 64 participants, as well as detailing the sports activities (basketball, cycling, and others) of 42 participants. This lack of data makes it challenging to identify the extent to which basketball referees are concerned about maintaining a good physical condition and whether the strategies used for this purpose vary according to their competitive status. In this context, we hypothesize that referees at the highest competitive level may be the ones who most frequently engage in physical fitness practices and undertake match-day exercise-based regimens.

Therefore, this study aimed to identify physical fitness practices among Spanish basketball referees at elite, sub-elite and regional competitive levels. The secondary objectives were: (a) to depict the demographic characteristics and officiating duties of Spanish basketball referees based on competition level, and (b) to describe their typical match-day exercise-based regimens depending on their competition level.

Materials and Methods

Design

A cross-sectional study was developed following the Strengthening the Reporting of Observational Studies in Epidemiology (STROBE) guidelines (von Elm et al., 2008).

Participants

We recruited 637 male and female basketball referees with experience in different Spanish leagues from 18 regional organizations via professional and personal contacts at the start of the 2021/22 season. The highest competitive level determined three categories: elite referees in the top professional men’s basketball league (Liga Endesa), sub-elite referees officiating in national non-professional leagues for men and women, and regional referees officiating in a region-wide or lower-level league.

Participants were informed of the study requirements and provided written informed consent according to the Declaration of Helsinki. The study protocol was approved by the University of Vigo Ethics Committee with code 01-1421.

Procedures

All regional referee organizations in Spain received invitations to participate in an online survey during the 2021/22 pre-season clinics, accompanied by a letter of invitation and written guidelines distributed. The credibility of a researcher, who is an active elite internationally ranked basketball referee, fostered trust among participants. The survey remained open for 11 weeks (26 August to 6 November 2021), with follow-up telephone reminders sent to each organization. Anonymous data was collected ensured participants’ privacy.

Questionnaire

According to previous investigations, it was deemed necessary to create a specific questionnaire to gather valuable data (Blagrove et al., 2020; Murphy, Mason & Goosey-Tolfrey, 2021). Four authors with expertise in basketball research and practice (three doctors and one PhD student in Physical Activity and Sports Sciences) designed an ad hoc questionnaire to collect data on physical fitness practices and match-day exercise-based regimens. The first author initiated the questionnaire development process by determining key areas of the questionnaire. A second author supervised this process. After debating how best to gather information, these two researchers developed an initial questionnaire including 24 items.

Subsequently, two other authors familiar with the needs of licensed referees by the International Basketball Federation (FIBA) reviewed the preliminary questionnaire and suggested shortening it for better readability. The questionnaire was revised accordingly.

Eventually, the fifth author contributed to refining the questionnaire. The final version, unanimously approved, comprised 24 closed and open-ended questions in Google Forms. It collected data on demographics, refereeing level, physical fitness practices, and match-day regimens, focusing on a regular pre-pandemic season to minimize confounding factors. The questionnaire began with informed consent and obtained personal and refereeing data, facilitating body mass index estimation using height and weight. It inquired about the specific season, competition types, weekly refereed match frequency, and categories officiated. The main section examined participants’ exercise habits, including regular exercise, personal coaching, exercise types, daily duration, and weekly frequency. It also explored activities during basketball competition: warm-up routines, time spent on routines, exercise routines during match breaks, re-warm-up (RW-U) activities, and reasons for performing RW-U or not. Additionally, it investigated participants’ stretching and joint mobility exercises. A comments section allowed for further insights, suggestions, or concerns.

Statistical and qualitative analysis

In this study, we utilized a descriptive cross-sectional survey design, which led us to primarily focus on descriptive analysis and data presentation. All statistical analyses were performed using the SPSS 15.0 (SPSS Inc., Chicago, IL, USA).

We calculated summary statistics for dichotomous or categorical variables and presented them as percentages. We also formulated contingency tables to identify systematic associations among the assessed variables. To analyze all variables, we employed the Chi-square test of independence ( χ2) or Fisher’s exact test when the contingency table was 2 × 2, setting the significance level at α = 0.05. The degrees of freedom (Df) for Chi-square test were calculated using the following formula: Df = (r − 1) (c − 1) where “r” is the number of rows, and “c” is the number of columns. Adjusted standardized residuals (CR) were applied to isolate the sources of variation among the groups (Haberman, 1973). Cramér’s V statistic was used to test the practical significance of these associations (values of 0.1 regarded as a small effect, 0.3 as a moderate effect, and 0.5 as large effects).

Drawing upon the investigative framework employed by McGowan et al. (2016) in their evaluation of swimming warm-up practices, we utilized a comparable approach for qualitative scrutiny in our study. Responses to our specific open-ended questions (namely, “If so, could you indicate the reason(s) for physically reactivating during the break between halves?” and “If not, could you indicate the reason(s) for not reactivating physically during the break between halves?”) were subjected to a systematic content analysis. Initially, a co-author applied an inductive method to distill the raw data (File S1) into common themes, which captured a range of motivations for warm-up practices and situational barriers. This phase also involved identifying intersecting themes and discarding responses that were either invalid or unclear.

The principal investigator then performed a deductive analysis for further examination, seeking empirical evidence to either confirm or refute the initially identified thematic structures. This review drew on studies of RW-U practices in team sports, particularly soccer and basketball. We referenced Towlson, Midgley & Lovell’s (2013) study to validate themes related to increased muscle temperature, mental readiness, and arousal for the first question. For the second question, themes such as lack of time, interference with players’ psychological and tactical preparation, and the physical readiness of players were examined, along with factors like match official instructions, prevention of player fatigue in later match stages, and the impact on nutritional strategies. González-Devesa et al. (2022a) study aided in validating themes concerning body temperature increase, mental preparation, and injury prevention for the first question, and time constraints for the second. Additionally, Anshel’s (1995) study, which defined relevant competencies for basketball referees, informed the theme of adequate mental and physical pregame preparation for the second question.

For the first question, we established three main motivational categories with their respective subthemes: “Activation” (including Arousal and Temperature regulation), “Mental readiness” (encompassing Concentration and focus, Decision-making improvement, and Stress management), and “Injury prevention”. In response to the second question, we identified three primary barriers: “Perceived lack of need” (with subthemes addressing Proper physical and mental preparation, Inherent self-sufficiency, the Unexplored concept of re-warm-up, and Absence of habit or routine), “Time constraints”, and “Competing needs” (which encapsulate Prioritizing technical aspects of refereeing, Rest and recovery strategies, and a combination of both these elements). Both sets of questions featured combinations of these categories, typically involving two or all three main categories, along with a distinct classification for invalid responses. A detailed breakdown of each main category and subcategory, including responses across regional, sub-elite, and elite referee levels, can be found in File S2.

Results

Out of 637 initial respondents, 628 referees (mean age: 30.0 ± 9.4 years; mean number of seasons serving as a referee: 8.9 ± 7.3; 15.4% female) were included after eliminating nine respondents with contradictory data. The final sample included 531 males and 97 females, aged 15 to 58 years (mean age 30.0 ± 9.4 years). Table 1 shows their demographic characteristics.

Table 1 Demographic characteristics and officiating duties of Spanish basketball referees in subgroups depending on competition level.

	Elite (n = 32)	Sub-elite (n = 109)	Regional (n = 487)	Total (n = 628)	p-value	Cramer’s V	Df	
Gender	0.059	0.095	2	
Male	93.8 (30) [1.5]	89.9 (98) [1.7]	82.8 (403) [−2.3]	84.6 (531)				
Female	6.3 (2) [−1.5]	10.1 (11) [−1.7]	17.2 (84) [2.3]	15.4 (97)				
Age (years)	0.000	0.270	6	
16–23	0 [−3.6]	2.8 (3) [−6.4]	35.1 (171) [7.7]	27.7 (174)				
24–28	15.6 (5) [−1.4]	28.4 (31) [0.6]	26.3 (128) [0.2]	26.1 (164)				
29–34	34.4 (11) [1.7]	45.9 (50) [6.5]	16.2 (79) [−6.8]	22.3 (140)				
>34	50 (16) [3.6]	22.9(25) [−0.3]	22.4 (109) [−1.6]	23.9 (150)				
Type of competitions	0.000	0.310	4	
Senior and youth	6.3 (2) [−4.7]	40.4 (44) [−1.5]	51.1 (249) [3.9]	47 (295)				
Senior	93.8 (30) [7.4]	59.6 (65) [6.4]	23.6 (115) [−9.7]	33.4 (210)				
Youth	0 [−2.9]	0 [−5.7]	25.3 (123) [6.7]	19.6 (123)				
Refereeing experience (years)	0.000	0.225	6	
≥3	0 [−3]	3.7 (4) [−4.9]	26.5 (129) [6.1]	21.2 (133)				
>3–6	12.5 (4) [−2.1]	25.7 (28) [−0.9]	31 (151) [1.9]	29.1 (183)				
>6–12	28.1 (9) [0.5]	36.7 (40) [3.2]	21.8 (106) [−3.1]	24.7 (155)				
>12	59.4 (19) [4.6]	33.9 (37) [2.4]	20.7 (101) [−4.6]	25 (157)				
Refereeing frequency (estimated matches per week during the regular season)	0.000	0.426	14	
1	53.1 (17) [10.6]	13.8 (15) [3.1]	2.3 (11) [−8.5]	6.8 (43)				
1–2	34.4 (11) [3.3]	34.9 (38) [6.6]	8.6 (42) [−7.8]	14.5 (91)				
2–3	12.5 (4) [−0.5]	21.1 (23) [1.6]	15 (73) [−1.2]	15.9 (100)				
+3	0 [−7.5]	30.3 (33) [−7.7]	74.1 (361) [11]	62.7 (394)				
Note:

Data are presented as % (n) (adjusted standardized residuals); bold values denote statistical significance (p < 0.05) among categories of referees; significant corrected typified residuals (−1.96 > rz > 1.96).

Female referees were underrepresented (6–17.2%) across the competitive level categories, precluding gender comparison. Elite referees, who were older and more experienced, officiated almost exclusively in senior competitions (93.8 %, χ2 = 120.945, Df = 4, CR = 7.4, p < 0.001) and none in more than 3 weekly matches ( χ2 = 228.435, Df = 14, CR = −7.5, p < 0.001). About 60% of sub-elite referees participated only in senior competitions (59.6%, χ2 = 120.945, Df = 4, CR = 6.4, p < 0.001), with 30.3% ( χ2 = 228.435, Df = 14, CR = −7.7, p < 0.001) involved in more than 3 weekly matches. All regional referees officiated in youth categories (25.3%, χ2 = 120.945, Df = 4, CR = 6.7, p < 0.001) and the majority had more than 3 weekly matches (74.1%, χ2 = 228.435, Df = 14, CR = 11, p < 0.001).

Estimated BMI mean values (23.8–24.2 kg·m−2) did not differ significantly across the three categories.

Physical fitness practices

Table 2 presents data on physical fitness practices. Almost all referees reported engaging in regular planned physical activity, with a few exceptions. Elite referees were most likely to hire personal trainers (31.3%, χ2 = 18.560, Df = 2, CR = 3.3, p < 0.001), followed by sub-elite and regional referees (20% and 12%, respectively).

Table 2 Physical activity habits of Spanish basketball referees in subgroups of different competition level.

	Elite (n = 32)	Sub-elite (n = 109)	Regional (n = 487)	Total (n = 628)	p-value	Cramer’s V	Df	
Regular physical activity	0.137	0.080	2	
Yes	100 (32) [1.1]	99.1(108) [1.6]	95.9 (467) [−2]	96.7 (607)				
No	0 [−1.1]	0.9 (1) [−1.6]	4.1 (20) [2]	3.3 (21)				
Personal trainer	0.000	0.172	2	
Yes	31.3 (10) [3.3]	19.3 (21) [2.4]	9.7 (47) [−3.9]	12.4 (78)				
No	68.8 (22) [−3.3]	80.7 (88) [−2.4]	90.3 (440 [3.9]	87.6 (550)				
Physical training frequency (days/week)	0.377	0.073	6	
1	0 [−1]	0.9 (1) [−1.2]	3.2 (15) [1.6]	2.6 (16)				
2–3	46.9 (15) [0.8]	38.5 (42) [−0.4]	40.1 (187) [−0.1]	40.2 (244)				
4–5	50 (16) [0.4]	52.3 (57) [1.2]	45.4 (212) [−1.3]	47 (285)				
6–7	3.1 (1) [−1.4]	8.3 (9) [−0.7]	11.2 (52) [1.4]	10.2 (82)				
Time spent in physical training practice (minutes/training session)	0.320	0.062	4	
<15	0 [−0.7]	0 [−1.4]	1.9 (9) [1.7]	1.5 (9)				
≥15–60	65.6 (21) [1.1]	60.6 (66) [1]	54.5 (254) [−1.5]	56.2 (341)				
>60	34.4 (11) [−0.9]	39.4 (43) [−0.7]	43.6 (203) [1.1]	42.3 (257)				
Type of exercises				
Aerobic	84.4 (27) [0.2]	85.3 (93) [0.6]	82.8 (384) [−0.7]	83.3 (504)	0.801	0.027	2	
Strength	90.6 (29) [2.2]	80.7 (88) [1.9]	70.7 (328) [−2.9]	73.6 (445)	0.008	0.126	2	
Speed	43.8 (14) [0.8]	41.3 (45) [1]	35.7 (165) [−1.3]	37.1 (224)	0.406	0.055	2	
Flexibility	41.9 (13) [2.5]	25.2 (27) [0.5]	21.9 (101) [−1.7]	23.5 (141)	0.035	0.106	2	
Other skills (agility, coordination, balance)	32.3 (10) [0.8]	26.6 (29) [0.1]	25.6 (118) [−0.5]	26.1 (157)	0.710	0.034	2	
Note:

Data are presented as % (n) (adjusted standardized residuals); bold values denote statistical significance (p < 0.05) among categories of referees; Significant corrected typified residuals (−1.96 > rz > 1.96).

No significant differences in weekly training days were observed among the referees’ categories (almost half of the participants exercised 4–5 days). Most participants trained for 15–60 min per session, regardless of their competitive level. Aerobic (83%) and muscle strengthening (73.6%) activities were common, whereas speed (36.9%) and flexibility (23.2%) were less frequent. Approximately 26% of the referees incorporated balance, agility, and coordination tasks in their training regimes. Elite referees focused more on strength ( χ2 = 9.640, Df = 2, CR = 2.2, p < 0.008) and flexibility ( χ2 = 6.680, Df = 2, CR = 2.5, p < 0.035) exercises than other categories.

Table 3 depicts the match-day exercise-based regimens. Warm-up activities were less common in the regional category (26.7%, χ2 = 12.027, Df = 2, CR = −3.3; p = 0.002); 80.4% of referees reported performing stretching and joint mobility activities prior to matches, while 50.6% did so after match. Such activities were less frequent during the half-time (26.1%) and between halves (2%). Among referees not conducting stretching and joint mobility activities (7%), the cited reasons were lack of need, time, habit, or knowledge. Sub-elite referees were significantly more prone to execute these activities post-match (70.6%, χ2 = 21.120, Df = 2, CR = 4.6, p < 0.001), while regional referees performed them less often (8.4%, χ2 = 6.705, Df = 2, CR = 2.6, p = 0.035).

Table 3 Spanish basketball referees’ match-day exercise-based regimens in subgroups of different competition level.

	Elite (n = 32)	Sub-elite (n = 109)	Regional (n = 487)	Total (n = 628)	p-value	Cramer’s V	Df	
Warm-up routine	0.002	0.138	2	
Yes	93.8 (30) [2.4]	84.4 (92) [2.2]	73.1 (356) [−3.3]	76.1 (478)				
No	6.3 (2) [−2.4]	15.6 (17) [−2.2]	26.9 (131) [3.3]	23.9 (150)				
Performing stretching/joint mobility exercises				
Before the match	84.4 (27) [0.6]	87.2 (95) [2]	78.6 (383) [−2.1]	80.4 (505)	0.109	0.084	2	
Between 1Q–2Q	6.2 (2) [1.7]	0 [−1.7]	2.3 (11) [0.6]	2.1 (13)	0.076	0.091	2	
During HT	18.8 (6) [−1]	22 (24) [−1.1]	27.5 (134) [1.5]	26.1 (164)	0.310	0.061	2	
Between 3Q–4Q	0 [−0.8]	0.9 (1) [−0.8]	2.3 (11) [1.2]	1.9 (12)	0.470	0.049	2	
After the match	46.9 (15) [−0.4]	70.6 (77) [4.6]	46.4 (226) [−3.9]	50.6 (318)	0.000	0.183	2	
Never	3.1 (1) [−0.9]	1.8 (2) [−2.3]	8.4 (41) [2.6]	7 (44)	0.035	0.103	2	
Re-warm-up routine	0.864	0.022	2	
Yes	31.3 (10) [0]	33 (36) [0.5]	30.2 (147) [−0.5]	30.9 (193)				
No	68.8 (22) [0]	67 (73) [−0.5]	69.8 (340) [0.5]	69.1 (434)				
Note:

Data are presented as % (n) (adjusted standardized residuals); bold values denote statistical significance (P < 0.05) among categories of referees; significant corrected typified residuals (−1.96 > rz > 1.96).

Analysis of referee responses on half-time re-warm-up practices

A total of 193 referees (30.7% of the total sample) across all competitive levels engaged in re-warm-up (RW-U) activities, with stretching and joint mobility being the most prevalent activities (86%), followed by jogging and running (34%). Strength (8.8%) and coordination and balance exercises (6.7%) were the least commonly used RW-U strategies (Table 3). The data presented in Fig. 1 pertain exclusively to these 193 respondents who engaged in RW-U activities. There were no significant differences in RW-U practices among different referee levels, except for “other” RW-U activities where elite (n = 3), sub-elite (n = 0), and regional referees (n = 8) selected this option.

Figure 1 Types of re-warm-up practices of Spanish basketball referees based upon competition level.

Bold values denote statistical significance (p < 0.05) among categories of referees.

Reasons for physical reactivation during half-time

In our analysis of referees’ reasons for physical reactivation during the half-time break, the “Activation” category emerged as the most frequently mentioned (for detailed responses and data, see File S2). This theme saw considerable responses from regional referees, especially in the subthemes of Arousal and Temperature regulation. “Mental readiness”, with a strong emphasis on Concentration and focus, was another prevalent theme, although less represented among elite referees. “Injury prevention” was recurrently identified, primarily by regional referees.

Regional referees were the most responsive across these themes and subthemes. Sub-elite referees showed a notable alignment with “Mental readiness”, particularly in Concentration and focus, mirroring the response pattern of regional referees. Elite referees, less represented in the dataset, had a distinct tendency toward “Activation”, citing both of its subthemes.

Reasons for not engaging in physical reactivation during half-time

In terms of reasons for not reactivating physically (for further details, refer to File S2), a considerable number of responses categorized under the “Perceived lack of need” theme came predominantly from regional referees, particularly highlighting Proper physical and mental preparation and Inherent self-sufficiency. “Competing needs” and “Time constraints” also emerged as key themes, with the former covering subthemes like Rest and recovery strategies and Prioritizing technical aspects of refereeing in a similar manner, and the latter primarily involving regional and sub-elite referees.

Across the participant spectrum, regional referees were the most responsive in all categories. Sub-elite referees notably aligned with Inherent self-sufficiency within “Perceived lack of need” and were responsive in Rest and recovery strategies under “Competing needs”. Elite referees, while fewer in number, showed a particular inclination toward Absence of habit or routine within “Perceived lack of need” and the combined aspects of “Competing needs”.

Discussion

Physical development and fitness programming for referees remain under-researched topics (Cunningham, Mergler & Wattie, 2022), even though they are of considerable interest to basketball governing bodies and practitioners. Our study revealed that Spanish basketball referees across all competitive levels consistently engage in weekly planned physical activity, with warm-up exercises being a common component. However, some sub-elite and regional referees do not include warm-ups in their routines. Moreover, the majority of the surveyed referees, approximately two-thirds, do not perform re-warm-up exercises.

In terms of the time commitment to physical activity, we found that basketball referees engaged in regularly planned physical activity, spending more than 15 min and up to 60 min in more than half of the cases, 2 to 5 days per week, irrespective of their competitive level. This figure is noteworthy because prior research suggests difficulties in balancing refereeing duties with exercise and other responsibilities (Anshel, 1995). Similarities exist with conventional training programs for sub-elite Spanish referees, who trained around 75 min, three times per week (Bayón et al., 2015); and a survey of 78 Brazilian referees at the regional level, where 82% of respondents undertook 6.3 ± 3.0 weekly hours of physical preparation (de Paula, da Cunha & Andreoli, 2021). Possible explanations for these results beyond referees enjoying the physical nature of their role (Warner, Tingle & Kellett, 2013) include the remarkable physical demands at both the elite and sub-elite levels (Nabli et al., 2016a; Leicht et al., 2020), which necessitate physical fitness training (Plessner & MacMahon, 2013). For instance, elite internationally ranked referees may follow FIBA’s conditioning program (FIBA, 2020a), while sub-elite (Bayón et al., 2015) and regional referees (Sobko et al., 2021) are keen to train to pass physical fitness tests for promotion (Inchauspe et al., 2020; FIBA, 2021).

One unanticipated finding was that despite elite basketball referees exercise for at least 30 min twice a week, fewer train more than 60 min, or more frequently (at least 6 days per week) than those in lower levels. Given the age profile of elite referees in our sample, which primarily falls within the 29–34 and over 34 age brackets, there may be a tendency to shift towards more strategic, less intensive exercise routines. This adaptation aligns with the anticipated physical decline typically starting in these age ranges (Castagna, Abt & D’Ottavio, 2007; Tittrington, 2021; Ridinger et al., 2017). Elite referees’ training schedules may also be affected by geographical, professional, or officiating constraints (de Paula, da Cunha & Andreoli, 2021). Concurrently, the vast majority of this elite group had up to two official matches weekly, while sub-elite and regional referees officiated more frequently. Considering that referees require a minimum of 10,000 h of pressurized decision-making experience to reach expertise (Mildenhall, 2014), we can assume that lower-level referees may emphasize match practice to enhance their performance and advance to the highest rank.

A second noteworthy result is that a significant fraction of Spanish basketball referees chose aerobic and strength conditioning over speed and flexibility training. Regarding the predominance of aerobic activities, basketball referees require adequate training to maintain cardiovascular fitness because of moderate cardiovascular load during competitions (Leicht, 2004; Suárez Iglesias et al., 2021) and compromised aerobic performance with age (Nabli et al., 2019), especially when they reach the peak of their careers (first-time candidates for a FIBA Referee License have to be 35 years of age or younger, with the age limit for the license being around 50–55 years) (FIBA, 2021). Furthermore, our sample typically engaged in strength training and sprints. Previous studies have identified similar training patterns among sub-elite (Bayón et al., 2015) and regional referees (de Paula, da Cunha & Andreoli, 2021).

Agility, coordination, and balance tasks are underrepresented in their exercise routines, despite their importance in performance (Inchauspe et al., 2020). Previous studies confirm these findings (Bayón et al., 2015; de Paula, da Cunha & Andreoli, 2021). However, this trend appears to be changing as FIBA includes agility (i.e., footwork) and balance training (i.e., proprioception abilities) in its digital manuals for internationally ranked referees (FIBA, 2020b).

To the best of our knowledge, this study is unique in its comparative analysis of basketball referees across three competitive levels. Although we found minimal differences in exercise types for physical fitness among the groups, elite basketball referees exhibited a higher emphasis on strength and flexibility training compared to their sub-elite and regional counterparts. This observation is likely due to the age-related physical decline, as half of the surveyed elite referees were over 34 years old, and physical performance impairment is expected to become more pronounced from age 30 onwards (Castagna, Abt & D’Ottavio, 2007). Consequently, devoting more training time to strength and flexibility exercises may help prevent injury and improve movement efficiency for these referees (Gregson, Weston & Helsen, 2006).

In addition, elite referees reported hiring personal trainers to a greater extent than sub-elite and regional referees. This decision might be in response to the challenges they face officiating elite athletes (Plessner & MacMahon, 2013), who are typically 15–20 years younger than the referees themselves (Tittrington, 2021) and have greater muscle mass compared to sub-elite players (Masanovic, Popovic & Bjelica, 2019). Consequently, the employment of personal trainers by elite referees emerges as a proactive strategy to maintain optimal fitness levels (Tittrington, 2021). Studies suggest that structured training under qualified personal trainers leads to more significant improvements in key fitness metrics, including lean body mass, compared to self-guided training (Storer et al., 2014). Furthermore, personal trainers enhance this effect by offering varied and personalized exercise programs, adapting diverse techniques to individual needs (Waryasz et al., 2016). This observation is aligned with our findings, as elite referees incorporate a more diverse range of fitness components in their training regimens than their counterparts. Additionally, it is worth noting that the financial resources available to elite referees, potentially including higher salaries or allocated budgets for personal training, may facilitate their greater use of personal trainers compared to sub-elite and regional referees.

When examining anthropometric measurements, our findings indicate that the mean body mass index values in our sample were lower than those documented in prior studies. Notably, these earlier studies revealed that elite male basketball referees often displayed above-normal weight ranges (Rupčić et al., 2011; Bonganha et al., 2013; Suárez Iglesias et al., 2021). The reason for this difference is unclear but could be due to the current trend of body image ideals for men valuing a muscular, low body fat physique (Robinson & Lewis, 2016). Documents from the Spanish Basketball Federation suggest training for hypertrophy to improve court presence (Federación Española de Baloncesto, 2022). It has been argued that standing firm and portraying confidence are key communication behaviors of the skillful referee (Plessner & MacMahon, 2013); therefore, being fit could enhance performance by increasing confidence (Morris & O’Connor, 2017) and court presence (Tittrington, 2021). Nonetheless, the interpretation of these findings remains speculative since the participants estimated their height and weight.

In the current study, many referees performed warm-up activities before a match, and this percentage increased with the level of competition. This underscores the importance of pre-match preparation for optimal performance across competitive levels (Anshel, 1995; de Paula, da Cunha & Andreoli, 2021). Elite referees seem to be more aware of establishing match-day routines, which may be due to the demands of top competition and access to standardized warm-up procedures (Suárez Iglesias et al., 2021). Altogether, developing education programs for sub-elite and regional referees to implement competition routines would benefit referee organizations. Some efforts in this direction are already underway through online media (Hrusa & Hrusova, 2020; de Paula, da Cunha & Andreoli, 2021; Sobko et al., 2021).

Conversely, half of our sample did not perform post-match stretching or joint mobility exercises, highlighting the community’s possible lack of knowledge. This situation is consistent with previous research showing that most referees do not consider post-match injury prevention strategies as ideal (de Paula, da Cunha & Andreoli, 2021). In our investigation, the proportion of sub-elite referees including these strategies was significantly higher than that of the elite and regional referees. Time constraints or a perceived lack of need may explain this discrepancy. Elite referees may have less time due to post-match conferences with their instructors (Hrusa & Hrusova, 2020); while regional referees, especially those between 16–23 years of age (35% of our sample), may be less familiar (Sobko et al., 2021) or interested in physical readiness (Hrusa & Hrusova, 2020) than other categories of referees.

Despite only a third of respondents engaging in RW-U activities, current literature, such as González-Devesa et al. (2021) and Silva et al. (2018), emphasizes their significance. No notable differences were found between referee categories in the adoption of these practices. Stretching and joint mobility exercises were prevalent, with jogging, running, and breathing techniques also commonly reported. Our respondents’ motivations for physical reactivation, predominantly citing “Activation” (particularly Arousal and Temperature regulation) and “Mental Readiness” (especially Concentration and focus), align with previous research by Towlson, Midgley & Lovell (2013) and González-Devesa et al. (2022a). These studies investigated half-time RW-U practices among elite soccer practitioners, and elite and sub-elite basketball practitioners. This parallel suggests a wide recognition of these RW-U aspects and their significance across different sports, roles, and competitive levels.

Our results present a marked contrast to González-Devesa et al. (2022a) in the context of “Injury prevention”. While their research observed minimal focus on this aspect, it emerged as a significant theme in ours, especially among regional referees. This disparity could stem from the different competitive levels and roles of participants between studies. Supporting this trend, de Paula, da Cunha & Andreoli (2021) noted a common occurrence of musculoskeletal overuse injuries in the lower limbs among Brazilian regional basketball referees, underscoring the relevance of injury prevention at this officiating level. In summary, these divergent findings integrate into the evolving debate in sports science regarding RW-U’s effectiveness in reducing injury rates, as highlighted in the recent discourse by Afonso et al. (2023).

The perception of RW-U as unnecessary was strongly held by regional and sub-elite referees who often considered their readiness to be sufficient during brief breaks. This perspective is partially supported by research indicating limited benefits from short, sport-specific RW-U interventions in young basketball players (González-Devesa et al., 2022b, 2023). For elite referees, the absence of established routines, rather than skepticism about its effectiveness, shaped their approach, suggesting opportunities for educational programs to instill beneficial RW-U practices and the need for further research in this area.

Additionally, “Competing Needs,” encompassing logistical, personal, technical, and tactical tasks, along with nutritional and rest considerations during breaks, were identified as a second major challenge, mirroring the findings of Towlson, Midgley & Lovell (2013). This illustrates the difficulty of effectively managing half-time periods across various sports levels (Afonso et al., 2023), more so when they are limited. In fact, referees cited “Time Constraints” as another barrier to perform RW-U strategies, aligning with challenges reported in English soccer and Spanish basketball (Towlson, Midgley & Lovell, 2013; González-Devesa et al., 2022a). Interestingly, elite referees, with breaks of up to 15 min, did not highlight time issues, in contrast to regional and sub-elite peers who struggled with breaks as brief as 3 to 10 min. Spatial limitations, while notable in literature (Towlson, Midgley & Lovell, 2013; González-Devesa et al., 2022a), were rarely a concern for our referees.

Despite the strengths of this study, including a large sample size and a broad spectrum of competitive levels, certain methodological limitations exist. These include using an untested “ad hoc” questionnaire and recall bias introduced by focusing on a pre-COVID-19 regular season. Social desirability response bias could also have affected the results, although pre-defined response options may mitigate this McEwan et al. (2020). The possibility that participants accounted for refereed matches as part of their regularly planned physical activity may have introduced report bias. Also, self-reported anthropometric measurements may be inaccurate, which limits further analysis. Additionally, the sample was predominantly male; hence, future research is required to establish the generalizability of the findings to both sexes.

Conclusions

Spanish basketball referees engage in regular physical activity and physical fitness practices, regardless of their competitive level. Although warm-up activities are common, they are not consistently performed by some sub-elite and regional referees, and re-warm-up routines are not widely adopted. The insights gained may help individual referees and their organizations optimize pre-match and in-match preparations, as well as enhance physical performance levels. This could contribute to establishing physical conditioning standards, particularly in light of the limited research on these topics to date. Future studies should explore optimal physical training approaches, the timing of each activity, and the factors influencing individual physical development among referees.

Supplemental Information

Supplemental Information 1 Raw data translated from Spanish to English with the responses of the participants to the questionnaire.

Click here for additional data file.

Supplemental Information 2 Raw data in the original Spanish with the responses of the participants to the questionnaire.

Click here for additional data file.

Supplemental Information 3 Textual and Visual Representation of Qualitative Analysis on Re-Warm-Up Practices Open-Ended Questions.

Click here for additional data file.

Supplemental Information 4 Raw data used for conducting statistical analyses with SPSS.

Click here for additional data file.

Supplemental Information 5 Online questionnaire used to gather data on the physical exercise habits in basketball referees translated from Spanish to English.

Click here for additional data file.

Supplemental Information 6 Online questionnaire used to gather data on the physical exercise habits in basketball referees in the original language (Spanish).

Click here for additional data file.

We thank all the participants taking the questionnaire and the following regional referee organisations for their involvement: Fed. Española de Baloncesto, Fed. Andaluza Baloncesto, Fed. Aragonesa Baloncesto, Fed. Baloncesto Pdo. Asturias, Fed. de Bàsquet de les Illes Balears, Fed. Canaria de Baloncesto, Fed. Cantabra Baloncesto, Fed. Baloncesto C. La Mancha, Fed. de Baloncesto de Castilla y León, Fed. Catalana de Basquetbol, Fed. Extremeña de Baloncesto, Fed. Gallega Baloncesto, Fed. Baloncesto de Madrid, Fed. Melillense Baloncesto, Fed. Baloncesto Región de Murcia, Fed. Navarra Baloncesto, Fed. Riojana Baloncesto, Fed. Vasca Baloncesto and Fed. Baloncesto Com. Valenciana. Besides, the authors would like to acknowledge the linguistic help provided by Amy Bonner (professional FIBA and WNBA referee).

Additional Information and Declarations

Competing Interests

Author Contributions

Human Ethics

Data Availability

Alejandro Vaquera and David Suárez-Iglesias serve as fitness coordinators for the International Basketball Federation (FIBA) and FIBA Europe, respectively. Alberto Sánchez-Sixto is responsible for physical preparation for the Spanish Basketball Federation Refereeing Area.

David Suárez-Iglesias conceived and designed the experiments, performed the experiments, analyzed the data, prepared figures and/or tables, authored or reviewed drafts of the article, and approved the final draft.

Daniel González-Devesa conceived and designed the experiments, performed the experiments, analyzed the data, prepared figures and/or tables, authored or reviewed drafts of the article, and approved the final draft.

Carlos Ayán conceived and designed the experiments, performed the experiments, analyzed the data, authored or reviewed drafts of the article, and approved the final draft.

Alberto Sánchez-Sixto performed the experiments, authored or reviewed drafts of the article, and approved the final draft.

Alejandro Vaquera conceived and designed the experiments, performed the experiments, authored or reviewed drafts of the article, and approved the final draft.

The following information was supplied relating to ethical approvals (i.e., approving body and any reference numbers):

The University of Vigo granted Ethical approval to carry out the study (code 01-1421).

The following information was supplied regarding data availability:

The raw data are available in the Supplemental Files.

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
