# Peer review of "Do you even exercise, ref? Exploring habits of Spanish basketball referees during practice and matches"

_PeerJ, doi:10.7717/peerj.16742_

## Round 0.1 · original submission · Major Revisions

Dear Authors

Your submission has been reviewed by two experts in the field of study. The comments of the reviewers are included at the bottom of this letter. We invite you to submit a revised version of the manuscript that addresses the points raised by the reviewers.

We look forward to receiving your revised manuscript.

Best regards

Yung-Sheng Chen, Ph.D.
Academic Editor

·

Basic reporting

Review
Original title: Do you even exercise, ref? Exploring habits of Spanish basketball referees during practice and matches

Dear Authors, I really appreciate Your work on this interesting and important topic. I believe that Your contribution should be helpful for officials and associations. Although I have some remarks.
Basic reporting
The structure of the paper is proper and the topic meets the criteria of PeerJ. The flow of the paper is good and English is professional. The literature cited is up-to-date.
Please adjust the text, put down to the following paragraph single letters like ‘a’ etc. and double check grammar and spelling.

Experimental design

Table 1
title of the table: Demographic characteristics and officiating duties of Spanish basketball referees in subgroups depending on competition level
Put ‘s of Cramer’s V together in one line
Refeering experience should be like:
Lower than or equal 3 years
Over 3 years up to 6 years
Over 6 years up to 11 years
Over 11 years
Refeering frequency:
Should be:
1
2
3
More than 3

Above remarks are very simple but they can be crucial for the results, statistics and conclusions basing on them. If this have impact on the numbers (for example where to put somebody who have 1 match per week – to category 1 or to category 1-2? Where to put somebody who have 3,5 year experience?) You need to give explanation to this – if this is only presentation of the results or does this have impact on the results??
Table 2
title - Physical activity habits of Spanish basketball referees in subgroups of different competition level
(-1.96 > r > 1.96) – please correct to (-1.96 > rz> 1.96)
Time spent in physical training practice (minutes/training session) – please correct to ‘on’
Same situation as with table 1 -> it should be
Less or equal 15(min)
Over 15(min) up to 30(min)
Over 30(min) up to 45(min)
Over 45(min) up to 60(min)
Over 60(min)
Please confirm that it has no impact on results

Table 3
Title - Spanish basketball referees' match-day exercise-based regimens in subgroups of different competition level
Figure 1 – change what you have to P  0.05
Please change color for regional to lighter so that the difference should be more visible
The vertical axis should be ‘percent of respondents per referees category’. Below the figure you can add that 100% is the n specific for each category.
Please add that as I suppose the p level is for all categories so it shows that there were/were not significant changes between categories but does NOT show between which – correct me if I am wrong.

Validity of the findings

Conclusions
There are wide differences between conclusions presented in the abstract and in the conclusions section.
Please be consistent here and rewrite the section and the abstract formulating concise conclusions which are the answers to the particular aims of the study.
F.e.
Line 36 – word still is not adequate here because You do not compare presence to the past in this study
Line 36-37 – you are talking about warm-up routines but you do not show data about warm-up. What you show is data about re-warm-up. Please be precise here. What is more – to state that referees really need warm-up routines you should prove that practicing warm-up routine gives some positive effects – f.e. decreasement in injury incidence or injury burden. BUT – this actual study does not present this kind of data.
Lines 38-40 – the same as above – your data does not prove this. Imagine – you should prove that stretching after the match decreases injury risk or injury burden to say that because you are talking about injury prevention here – so what prevention – injuries after the match? Or overuse injuries ?
Lines 40-41 – the same as above – your data has nothing to say about relationship of re-warm-up and performance enhancement.

One of the main limitations of this study is the question IF the respondents include match refeering into their training frequency (how much times a week) and training time (how long a day).
Some of them DID include this, some DID NOT. Imagine 2 persons who have 3 matches (of 40 minutes)/week and 1 training session of 30min – it means person A can report 4 times a week frequency and 150minutes of activity, if person A includes warm-up and time between Q + cool down it can give over 250minutes (120 match time, 30 training session, ~100min 3xwarmups+cool downs).
Person B who does not count in neither matches, warm-ups nor cool downs can report 1 time a week of 30minutes and that’s all. And that is the main problem of this study.

Additional comments

The methodological problem is very simple, but it is crucial. Lets give the Authors the chance to reply to the above remarks and after that reconsider decision.

·

Basic reporting

Some parts of the manuscript need revision to improve quality. Literature references are sufficient.

Experimental design

Methods section needs revision for coherence. One crucial issue is inclusion of females which might have affected the results (I suggest comparing the results with vs without females for confirmation, and deciding whether to include/exclude females). Qualitative method was mentioned but was not used.

Validity of the findings

Statistical methods were robust, however Figure 1 depicted p-values from a smaller sample size, needing clarification.

Additional comments

Ln 36-41: The results section seemed recommendations. Revise by indicating the main findings of the study.
Ln 49-60: Unclear topic sentence (suggestion: Fitness plays a crucial role in unbiased judgements among basketball officiating). Some items (e.g. body composition can be removed).
Ln 71-73: This should be located at last paragraph of the introduction.
Ln 95-97: merge above.
Ln 121-125: This should be under the procedure subheading. Paraphrase to improve quality.
Ln 100-119: Under ‘Procedures’ include a measure/questionnaire and place these here. Paraphrase to make it succinct. Describe the questionnaire items you used in the study.
Also include ‘BMI’ under measure and provide how BMI was estimated.
Ln 142-143: You can remove this as this hasn’t been presented in the results section.
Ln 210-225: Highlight the findings of the study in first paragraph of the ‘Discussion’. Move the rest to another paragraph.
Ln 255-261: improve topic sentence (e.g. Elite referees also demonstrated ..)
Ln 263-271: Is it also possible that elite trainers have greater salaries or allotted budget for personal training vs others. You can also include this perspective.
Ln 273-274: improve topic sentence.
Figure 1. Sample size was smaller than what is presented. Revise accordingly.

---

## Round 0.2 · Minor Revisions

Dear Authors. Your revision has been reviwed by the reviewer Please find the annotated manuscript addressed by the reviewer 2. The qualitatitive anlaysis and result intepretation are required to be rephased in the revision.

·

Basic reporting

Dear Atuhors, Editor and Reviewer
The Authors did lots of work improving this manus according to reviewers' remarks and suggestions.
All of my comments have been answered accordingly and pointed issues have been improved.
In my opinion the manuscript is now ready to publish.

I would like to thank Authors, Editor and other reviewer for succesful cooperation and wish all of You good luck in Your future work.
all the best
Jarosław Muracki, PhD

Experimental design

all my suggestions were improved

Validity of the findings

all my suggestions were improved

Additional comments

all my suggestions were improved

·

Basic reporting

Some parts of the manuscript need revision to improve quality. Literature references are sufficient.

Experimental design

The authors need to expand the qualitative analysis process.

Validity of the findings

All underlying data have been provided; they are robust, statistically sound, & controlled. Qualitative results need to be expanded.

Additional comments

Although the efforts in revision are commendable, the authors must provide additional information on qualitative analysis/results and integrate the suggestions provided in the document.

---

## Round 0.3 · Minor Revisions

Dear Authors. The reviewer has a concern regarding to the qualitatitive anlaysis of your study. Please addressing the issues raised by the reviewer for next round of revision.

·

Basic reporting

Some parts of the manuscript need revision to improve quality. Literature references are sufficient.

Experimental design

The authors need to expand the qualitative analysis process.

Validity of the findings

All underlying data have been provided; they are robust, statistically sound, & controlled. Qualitative results need to be expanded.

Additional comments

The revised manuscript looks better but some changes are suggested for improvement. The authors should also consider inserting the qualitative inference in the discussion.

Ln 67-68: insert reference. Paraphrase and merge with Ln 68-71.
Ln 238: change reveals to revealed.
Ln 242: RW-U to full text description and thereafter in the discussion part.
Ln 260-262: Confusing. Revise.
Ln 273-274: This sentence is out of context since you are focusing on aerobic and strength and conditioning. I suggest you write a separate paragraph with this topic sentence.
Ln 300-301: Unclear. Revise.
Ln 310: check spacing

---

## Round 0.4 · accepted · Accept

Dear Authors

I would like to express my grateful thanks for your patience and efforts to improve the quality of the manuscript. Your submission is now endorsed for acceptance of publication in PeerJ. Congratulations!!! Before the publication, please check the abbreviations for consistency.

Best Regards

Yung-Sheng Chen, Ph.D.
Academic Editor